# Production, reproduction and some adaptation characteristics of Boran cattle breed under changing climate: A systematic review and meta-analysis

**Merga Bayssa** *, **Sintayehu Yigrem, Simret Betsha, Adugna Tolera**

College of Agriculture, Hawassa University, Hawassa, Ethiopia

* mergabayssa@yahoo.com

## Abstract

### Introduction

Climate change affects livestock production and productivity, which could threaten livestock-based food security in pastoral and agro-pastoral production systems of the tropics and sub-tropics. Boran cattle breed is one of the hardiest Zebu cattle reared by Borana Oromo pastoralists for milk and meat production. However, there is limited comprensive information on production, reproduction and adaption traits of the Boran cattle in Ethiopia. Thus, this paper aims to compile the main production, reproduction and some adaptation traits of Boran cattle based on systematic review and meta-analysis of peer reviewed published and unpublished literature.

### Methodology

A combination of systematic review and meta-analysis based on PRISMA guideline was employed. Accordingly, out of 646 recorded articles identified through database searching, 64 were found to be eligible for production, reproduction and adaptation characteristics of the Boran cattle, 28 articles were included in qualitative systematic review while 36 articles were used for quantitative meta-analysis.

### Result

The Boran cattle breed has the ability to survive, produce and reproduce under high ambient temperature, utilize low quality forage resources, and resist water shortage or long watering intervals and tick infestations. The review revealed that the breed employs various adaptation responses (morphological, physiological, biochemical, metabolic, cellular and molecular responses) to cope with harsh environmental conditions including climate change, rangeland degradation, seasonal feed and water shortages and high incidences of tick infestations. The meta-analysis using a random-effects model allowed provision of pooled estimates of heritability and genetic correlations for reproduction and production traits, which could be used to solve genetic prediction equations under a population level in

**Data Availability Statement:** All relevant data are within the manuscript and its Supporting Information files.

**Funding:** The German Academic Exchange Service (DAAD) and the Federal Ministry for Economic Cooperation and Development (BMZ) financed the research under CLIFOOD.

**Competing interests:** The authors have declared that no competing interests exist.

purebred Boran cattle. In addition, heritability and genetic-correlation estimates found in the present study suggest that there is high genetic variability for most traits in Boran cattle, and that genetic progress is possible for all studied traits in this breed.

## Conclusion

The Boran cattle breed has the ability to survive, produce and reproduce under high ambient temperature, utilize low quality forage resources, and resist water shortage or long watering intervals and tick infestations. However, currently there are several challenges such as recurrent droughts, pasture deterioration and lack of systematic selection and breeding programs that play to undermine the realization of the potential of the breed. Thus, we recommend systematic selection for enhancing the reproductive and production performances without compromising the adaptation traits of the breed coupled with improved management of rangelands.

## 1. Introduction

Most developing countries have been facing severe poverty due to the devastating effects of the climate change on the agriculture and food production systems. Like other agricultural sectors, climate change adversely impacts the livestock sector, of which cattle are one of the most susceptible species [1]. Recurrent drought, feed and water scarcity and endemic tropical diseases are the major factors which negatively influence livestock production in the east African rangelands [2]. Hence, it is critical to identify agro-ecological specific climate resilient adaptive animals to sustain livestock production. Ethiopia is endowed with diverse livestock genetic resources due to variable agro-ecology and geographical proximity to the centre of livestock domestication [3].

The Boran cattle breed, categorized as *Bos indicus* (humped Large East African Shorthorn Zebu type) is one of the local cattle breeds reared by the Borana Oromo pastoralists in the Borana rangelands of southern Ethiopia for meat and milk production. The animals possess several adaptive mechanisms which are helpful for their survival in harsh environmental conditions. Adaptive characteristics to warm climates encompass a wide range of physiological functions, behavioural and morphological attributes. The Boran cattle have special merits of surviving, producing and reproducing under high ambient temperature, seasonal fluctuations in feed supply and quality, water shortage and high diseases incidence including tick infestations [4, 5]. In addition, they are noted for their docility, high fertility, early maturity and ability to walk long distances in search of feed and water than most other *B. indicus* breeds [2].

Boran cattle are very versatile and well adapted to arid and semi-arid environments. The cows are very efficient converters of pasture forage into body fat deposits, which are later, mobilized during periods of feed scarcity and lactation. The cows therefore hardly lose body conditions during lactation or slight droughts [6]. However, declining feed availability and quality due to increasing population pressure, land use change, rangeland degradation, bush encroachment and climate change are posing serious challenges to livestock productivity in the Borana lowlands. In addition, Ethiopian Boran cattle breed is under threat due to several factors such as recurrent drought, lack of systematic selection and breeding programs and dilution with other breeds [7]. The breed is under threat from genetic erosion due to the admixture of other breeds that are used for restocking after drought [1]. However, there is limited comprensive information on production, reproduction and adaption traits of Boran cattle

breed in Ethiopia. Therefore, the current review provides an overview of efforts made to summarize production, reproduction and adaptation traits of Boran cattle and major challenges for Boran cattle breed improvement and conservation programmes in the pastoral and agro-pastoral production systems of Borana lowlands.

## 2. Materials and methods

### 2.1. Scope of the study and evaluated traits

We used a mixture of systematic review and meta analysis for synthesis of this review. To do this, we used published papers, unpublished literatures and case reports as our data, which enabled us to pool non-standardized and qualitative information [8] for adaptation traits while for production and reproduction parameters [5] quantitative parameters were pooled to obtain estimated mean values. The implementation of the systematic and meta-reviews followed the Preferred Reporting Items for Systematic Reviews and Meta-Analyses (PRISMA) guideline [9] and the checklist used to ensure inclusion of relevant information (Fig 1). These include (1) characterization of the research question, namely "what are main production, reproduction and adaptation characteristics of Ethiopian Boran cattle breed as compared to other zebu cattle in the country?", (2) web-based literature retrieval of relevant journals and case studies (3) selection of relevant studies by scanning the abstracts and titles of individual papers (4) abstraction of information from selected sets of final articles (5) determination of quality of the information available in these articles (6) evaluation of the extent of heterogeneity of the articles [9, 10]. The outcomes of interest were production, reproduction and adaptation characteristics of Boran cattle breed and the questions addressed were the following: (i) number, diversity, and scope of studies carried out so far, (ii) heterogeneities of estimates, (iii) pooled weighted mean estimates, and (iv) factors associated with the production, reproduction and adaptation characteristics of Boran cattle breed.

### 2.2. Study identification and data extraction

We used databases from AGORA, SCOPUS, Google Scholar, Google web, PubMed, Science Direct, CAB direct, African Journals online (AJOL) and lists of references of articles from peer reviewed publications, unpublished literatures and case reports. The language of publications was restricted to English. The key words utilized for electronic searches were: 'growth AND meat production AND Boran cattle', Milk production AND Boran cattle breed', 'Reproduction AND Boran cattle breed and Adaptation AND Boran cattle breed'. The literature searches were conducted from June 1, 2019 to December 15, 2019 by two independent reviewers. Eligible studies were selected by using inclusion and exclusion criteria. A study was eligible if it fulfilled the following criteria: (i) full text and published in English (ii) carried out on Boran cattle breed and published before December 15, 2019, (iii) cross-sectional/ longitudinal/case control studies, (iv) relevant response variables of production, reproduction and adaptation traits. In the succeeding steps, studies carried out before 1985 and unavailable papers were excluded, and full text reports were screened for eligibility. Exclusion of screened reports was done by the following criteria: duplication (articles/data), only conducted on cross breeds (to reduce bias/ method related heterogeneity), sample size (<50 records in animal studies for production and reproduction traits), and inconsistent data (data incoherent within a table or tables or in the narrative sections, and could not be figured out).

Fig 1 presents a flow diagram of the search and selection of studies. A total of 646 reports written in English were identified, of which 618 were original papers, 20 unpublished literatures and 8 review papers (5 narratives and 3 systematic review on adaptation characteristics). Five hundred ninety-eight original reports were selected after the removal of duplicates. At

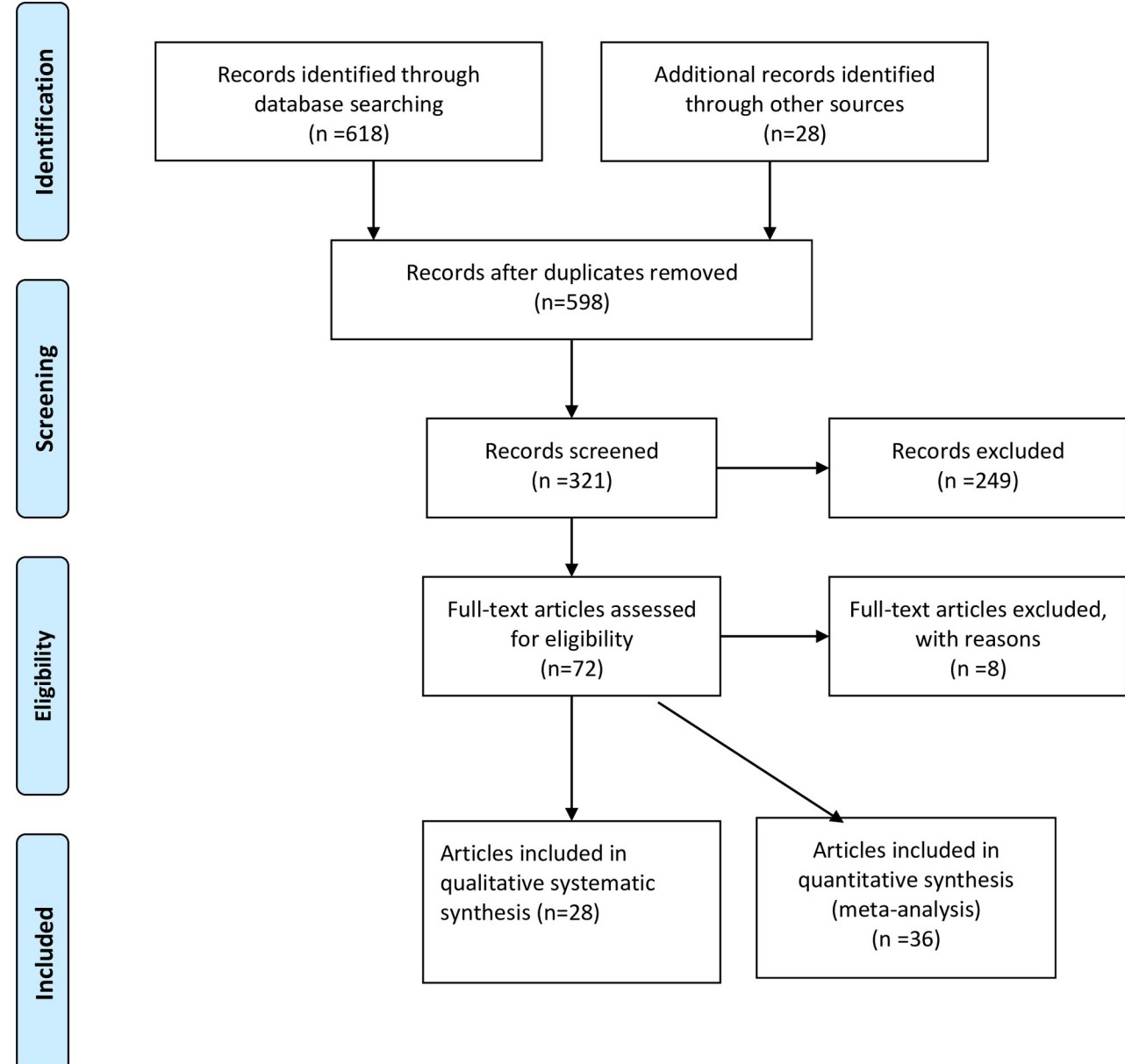

*Adapted From:* Moher D, Liberati A, Tetzlaff J, Altman DG, The PRISMA Group (2009). *P*referred *R*eporting *I*tems for *S*ystematic Reviews and *M*eta-*A*nalyses: The PRISMA Statement. PLoS Med 6(7): e1000097. doi:10.1371/journal.pmed1000097 with some modification

**For more information, visit www.prisma-statement.org.**

**Fig 1. Flow diagram of manuscript retrieval and study selection.**

screening stage, 321 papers were selected based on the selection criteria of their abstracts, of which 72 articles were subjected to full text assessment for validity. Finally, 64 articles were found to be eligible for adaptation, production and reproduction characteristics for full text extraction whereas 8 articles were excluded for failing to fulfil the selection criteria. Out of these studies, 28 articles were included in qualitative systematic synthesis of adaptation characteristics while 36 articles were used for quantitative meta-reviews of production and reproduction characteristics. The number, diversity and scope of studies carried out so far are inadequate for production and reproduction traits while both data from specific adaptation traits of Boran cattle breed and non-specific adaptation characteristics of other zebu cattle breeds were utilized for synthesis of the systematic reviews on adaptation characteristics due to crtical shortages of literatures on specific adaptation traits of Boran cattle while we conducted literature reviews.

The following data were extracted from eligible studies: first author, year of publication, year of study, country, sample size (herd size/records), sampling methods (probability/nonprobability based), breed (Boran, crosses/other zebu breeds) and study traits (production, reproduction and adaptation traits). For adaptation traits relevant information on morphological, behavioural, physiological, neuro-endocrine, blood biochemical and metabolic and molecular responses related to heat stress, water and feed scarcity and challenges of tropical diseases were gathered. For reproduction and production traits information on direct and maternal heritability, genetic correlations among traits and published standard errors for the relevant parameter estimates were included. Other information recorded were journal and database name, number of records, years of data collection, phenotypic mean and standard deviation, and the used estimation method (REML or Bayesian). When the same estimate was reported in different publications, based on the same database, only the most recent publication was included in the analysis. Besides that, meta-analysis was executed only for traits in which the estimates were based on at least two different databases, to minimize the possible impact of non-independence among articles. For articles in which the standard error for the heritability or correlation estimates were not reported, approximated standard errors were derived by using the combined-variance method [11] which is given as:

$$SE_{ij} = \sqrt{\left(\left(\sum\nolimits_{k=1}^{k} s^2 ik\, n^2\, ik / \sum\nolimits_{k=1}^{k} nik\right)/nij\right)}$$

Where $SE_{ij}$ is the predicted standard error for published parameters estimate for the i[th] trait in the j[th] article that has not reported the standard error, $s_{ik}$ is the published standard error for the parameter estimate for the i[th] trait in k[th] article that has reported the standard error, *nik* is the number of used records to predict the published parameter estimate for the i[th] trait in the k[th] article that has reported standard error, and $n_{ij}$ is the number of used records to predict the published parameter estimate for the j[th] article that has not recorded standard error. Since published correlations don't have normal distribution, they were converted to the Fisher's Z scale and all analyses are performed using transformed values. The results, such as the estimated parameter and its confidence interval, would then be back transformed to correlation for presentation [12]. The approximate normal scale based on Fisher's Z transformation [12, 13] is as follows:

$Z_{ij}$ = 0.5[ln(1+$r_{gij}$)-(ln(1-$r_{gij}$)], where $r_{gij}$ is the published genetic correlation estimate for the ith trait in the jth article. To return to the original scale, the following equation [12] was used:

$r^*_{gij}$ = ($e^{2zij}$-1)/($e^{2zij}$+1), where rgij is the re-transformed genetic correlation for the i[th] trait in the j[th] article and $Z_{ij}$ is the Fisher's Z transformation.

## 2.3. Quality assessment and quality control

After reviewing the papers, extracting data and completing summary tables, the methodological quality was assessed using the Mixed-Methods Assessment Tool by two reviewers. Reviewers met then to discuss discrepancies as well as the overall strengths and limitations of the studies. Discrepancies were overcome through consensus. Within and across study analysis and synthesis of the findings sections was undertaken using thematic synthesis. Studies were included in the final review if they scored 50% or greater, according to the scoring scheme [12]. The items were graded as high, moderate, low and very low categories. In general, those evidences with high and moderate classes were included for this particular review for studies considered for both systematic and meta-reviews.

In addition, box plots weighted by the number of records were constructed for each trait to identify potential outliers. To ensure the reliability of the meta-analysis, and avoid the estimation of biased estimates, a minimum number of articles required for each $i^{th}$ trait were obtained through the following relative standard error [14]: $RSE_i = ((s_i/\sqrt{n_i})/X_i)*100$, where $RSE_i$ is the relative standard error, $s_i$ is the standard deviation estimated from the published parameter estimates for the $i^{th}$ trait, $n_i$ is the number of articles that have reported parameter estimates for the $i^{th}$ trait, and $X_i$ is the average of parameter estimates for the $i^{th}$ trait. Traits with a RSE higher than 25% were discarded.

For reproduction and production traits considered in this review, weighted box plots were constructed by using $RSE_i$ values of each trait to identify potential outlier and to ensure the reliability of the data for the meta-analysis as this method was recommended as effective tool to evaluate reviews with a smaller number of articles and higher CV in published parameters [15]. Based on suggestion of [14], none of the traits recorded revealed to have $RSE_i$ higher than 25% (Fig 2) and thus, the available data on the important traits were considered.

The wide variability among estimates generated by the different studies for each trait observed by the higher coefficient of variation (CV) values in the traits considered in the reviews showed that there is large variation in the traits both between studies and within studies thus, random effect model is the methods of choice for the meta-analysis of the traits [12]. In addition, meta-analysis based on random-effect model is relevant for inferences at the population level [16–18].

For heritability estimates of reproduction and production traits, only direct heritability was considered excluding of maternal heritability and additive effect of environment due to limited data records on the considered parameters. Due to small numbers of articles reporting the reproduction and production heritability estimates and high variation among them, we were unable to investigate possible correlations in the present review; this analysis might be possible in the future when more studies become available on the Boran cattle breed.

## 2.4. Data analysis

Qualitative information on adaptation traits was summarized and tabulated for synthesis while production and reproduction data were analysed by using meta for package of R software [19].

**2.4.1. Phenotypic trait.** Means and standard deviations were calculated for all traits by using the sample sizes as weights. The total number of records for each phenotypic trait was calculated as the sum of number of records in each article that reported the trait. The coefficient of variation in percentage (CV (%)) for each $i^{th}$ trait was calculated as follows: CV (%) = $s_i/X_i *100$, where $s_i$ is the standard deviation for the $i^{th}$ trait, and $X_i$ is the trait mean.

**2.4.2. Heritability and genetic correlation.** Meta-analysis was performed on the basis of a random-effects model [12], in which the parameter estimates for all traits were analysed by assuming independence and normality.

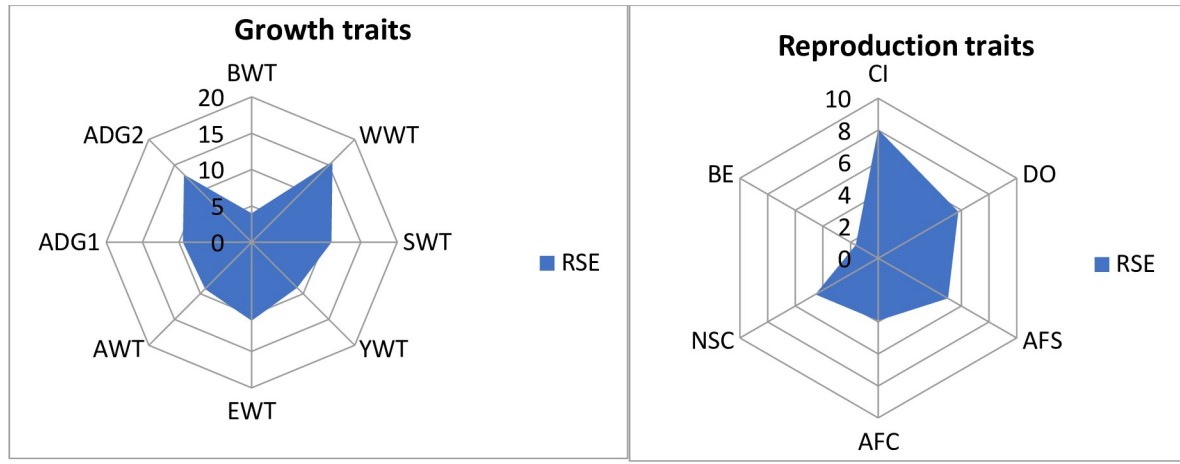

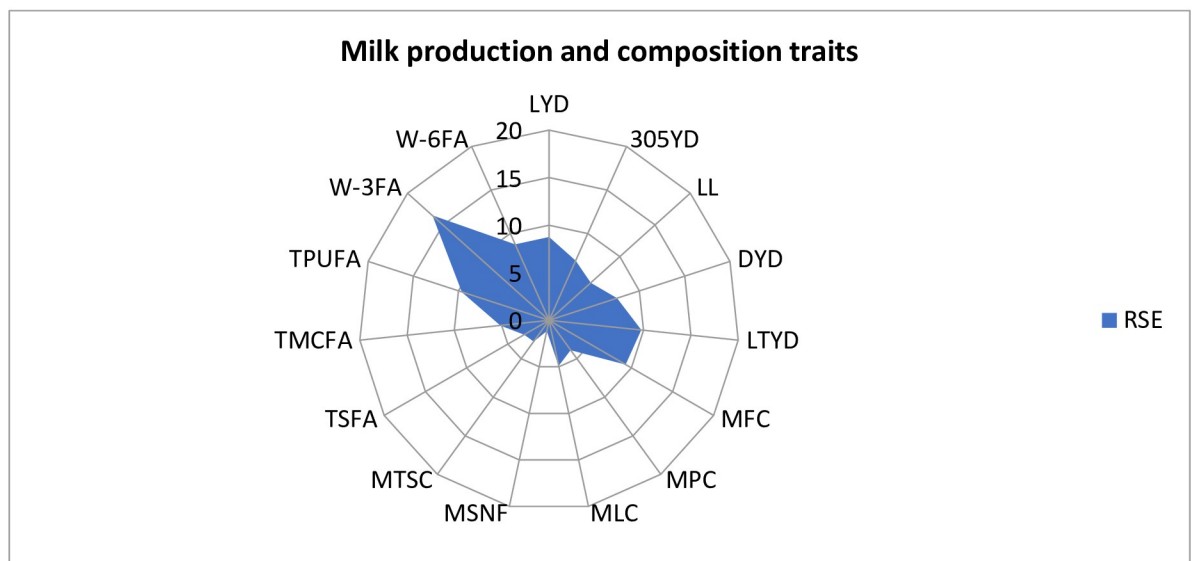

**Key**: *RSE=Residual stand error, BWT=Birth body weight, SWT=six month body weight, YWT= yearling bodyweight, EWT=Eighteen months body weight, AWT=anual body weight, ADG1=avaerage daily gain from birth to weaning, ADG2= avaerage daily gain from weaning to adult age, AFC=Age at first calving, NSC=number of sevice per conception, DO=days open, AFS=Age at first service, CI=calving interval, BE= , LYD=Lactation yield, 305YD=305 days milk yield, LL= lactation length, DYD=daily milk yield, MFC=milk fat content, MPC=Milk protein , content, MLC= milk lactose content, MSNF=Milk sold-not- fat content, MTSC=milk total solid content, TSCFA=total shotr chain faty acid, TMCFA=total medium chain fatty acid content, TPUFA=total polyunsaturated fatty acid content, w-3 FA=Omega three fatty acid and W-6 Fa=Omega six fatty acid*

**Fig 2. Box plots for relative standard errors of production and reproduction traits of Boran cattle calculated from the included search data.**

These assumptions were investigated for each trait by using the Box–Pierce and Shapiro–Wilk tests, available respectively in the *box.test* and *shapiro.test* functions of the R software [19]. The *meta for package* [20] available in the R software [19] was used to fit the following random-effects model for each trait: $\tilde{\theta}_j = \tilde{\theta} + u_j + e_j$ where $\tilde{\theta}j$ is the published parameter

estimate in the j[th] article, $\tilde{\theta}$ is the weighted population parameter mean, uj the among study component of the deviation from mean, assumed as $u_i \sim (N(0, \tau^2)$, where $\tau^2$ is the variance representing the amount heterogeneity among the studies, $e_j$ is with-in study component due to sampling error in the parameter estimate in the jth article, assumed as $e_j \sim (N(0, \sigma_e^2)$, where $\sigma_e^2$ is the with-in variance. The $I^2$ index [21] used to quantify the degree of heterogeneity among studies ($\tau^2$) for each trait can be described as follows: $I^2 = ((Q-df)/Q)$ x100,

Where Q is the Q statistics [22] given by

$$Q = \sum_{k=1}^{k} wi(\tilde{\theta}j - \tilde{\theta})^2$$

Where $w_j$ is the parameter estimate weight (as the inverse of published sampling variance for the parameter, 1/v) in the j[th] article; $\tilde{\theta}$rtand $\tilde{\theta}$ were defined above in the random-effects model; and the d.f. is the degrees of freedom (J-1, where J is the number of used articles) of a chi-squared distribution assumed for expected Q value on the assumption that $\tau^2 = 0$. Basically, $I^2$ values estimated as 25%, 50% and 75% might be considered as low, moderate and high heterogeneity respectively [21]. The 95% lower and upper limits for the estimated parameter would be computed respectively for each trait as follows:

$$LL\tilde{\theta}- = \tilde{\theta} - 1.96 \text{ x } SE\tilde{\theta} - \text{and } UL\tilde{\theta}- = \tilde{\theta} + 1.96 \text{ x } SE\tilde{\theta},$$

Where SE$\tilde{\theta}$is the predicted standard error for the estimated parameter θ, given by

$$SE\tilde{\theta} = (1/\sum_{J=1}^{J} wj.)$$

## 3. Results and discussion

### 3.1. Adaptation characteristics of Boran cattle breed

**3.1.1. Adaptation to harsh environmental conditions.** The summary of results of the systematic review of adaptation characteristics of Boran and other zebu cattle breeds were given under this subsection. The focus of the 28 articles reviewed were split into seven categories of adaptation traits, which included morphological (11), behavioural (8), physiological (7), neuro-endocrine (6), blood biochemical (7), metabolic (4) cellular and molecular (9) adaptation traits of Boran and other zebu cattle breeds in the tropics.

Boran cattle have tremendous adaptive capacity to harsh environment through specific morphological and physiological characteristics [6, 7, 23, 24]. On the other hand, heritability ($h^2$) estimates of adaptation traits were not determined for Boran cattle in this study due to scarcity of published data for the traits. In some tropical cattle breeds heritability estimates for heat tolerance ($h^2 = 0.18$–$0.75$) and tick resistance ($h^2 = 0.15$–$0.44$) showed a wide range of variabilities [25–27]. Heritabilities for resistance to nematode ($h^2 = 30$–$0.38$) and for trypanotolerance ($h^2 = 0.30$–$0.88$) vaied from medium to high heritabilities [26, 28]. Available reports [1, 26, 29] show that Boran cattle have adaptive traits to heat stress, external and internal parasites, and as well as to some blood parasites such as trypanssomiasis and east cost fever. These adaptive mechanisms of the Boran cattle to the harsh environmental conditions could be due to morphological, behavioural, physiological, neuro-endocrine, blood biochemical, metabolic, and molecular and cellular responses which are combined to promote survival and production in a specific environment [30].

**3.1.2. Morphological adaptation.** Indigenous cattle breeds adapted to arid and semi-arid regions possess special morphological features, such as skin coat, which helps to provide better protection from direct solar radiation [31]. Phenotypically, the typical Boran cattle breed (Qorti subtype) have white color, large dewlap and well-developed hump. But mostly they

have light grey or fawn colour with black or dark brown shading on the neck, head, shoulders and hindquarters, shorter and more pendulous sheath, well developed udder, long legs, wide ears and large dewlap, small humps, a short tail and erect and short horns with dominantly thick base [24, 32].

Phenotypic characteristics are resultant of adaptation mechanisms of the breed. Coat colour was one of the important morphological traits which impart adaptive ability to heat stressed livestock. For instance, white/ light grey coloured coats are helpful in thermo-regulation ability in tropical regions as it reflects 50% to 60% of direct solar radiation compared with the dark-coloured animals [1, 33]. Light or white-coloured coated animals are recognized as being advantageous in hot tropical regions. On the other hand, highly pigmented skin protects the deep tissues from direct short wave UV radiation by blocking its penetration. In addition, skin hair coat length, thickness and hair density also affect the adaptive nature of animals in tropical regions, where short hair, thin skin and fewer hair follicles per unit area are directly linked to higher adaptability to warm climate [34, 35]. Indigenous cattle breeds are more adapted to high temperatures, high solar radiation and dry conditions than exotic cattle breed due to their high skin pore density which allows them to successfully regulate their body temperature [36]. The Boran cattle also exhibit long legs to trek long distance (usually more than 60 km/day) in search of feed and water [24, 29] and wide ear and large dewlap to increase surface area for metabolic heat dissipation [7, 37].

In addition, the Boran cattle have higher level of resistance to biting insects and tick infestation due to: (1) A highly sensitive and motile skin with thick and well-developed layer of subcutaneous tissue causes the muscles beneath their skin to contract and move in reaction to insects landing on and biting them. This enables them to vigorously shake off external pests, (2) A very short hair coat makes it difficult for insects to attach onto hides of Boran cattle (3) A waxy and oily secretion from the skins of the Boran cattle makes them less desirable host for ticks and flies and (4) A long tail with a big well-formed twitch used to ride off vector flies for eye infection from Boran cattle [34, 38]. It was also reported that Boran cattle have prominent, protective eyebrows and long eyelashes which protect their eyes from bright sunlight, dust and other irritants factors that predispose cattle to pinkeye infection [1]. The Boran cattle (Kenyan Boran) tend to present some resistance to trypanosomal infection in the tsetse-infested regions of the east coast of North East Kenya [2]. Borana cattle both bulls and cows usually have horns and their horns are relatively smaller in size which was believed to have lower energy requirement for maintenance and have probably more adaptive capacity as compared to cattle breeds with large horns [1]. Morphological traits in livestock are highly important from the adaptation point of view, as they directly influence the heat exchange mechanisms between the animal and the surrounding environment [33].

**3.1.3. Behavioural adaptation.** In an effort to adapt to varying environmental conditions, animals exhibit several behavioural responses. The most important behavioural responses studied in tropical cattle include: shade seeking, standing time, feed intake, defecating and urinating frequency, water intake and frequency [39]. Tropical indigenous cattle breeds were observed to be highly adapted to direct heat stress, spending more time for grazing than resting in shade. Zebu cattle have the ability to adapt their grazing behaviour in response to restricted grazing time or when no grazing is allowed during nights [40–43]. Several studies show that different Zebu breeds spend more than 4% of their pasture time resting which varies between 4.4% and 10.12% [41, 42]. In a study involving Boran cattle, the resting time is as low as 2% [40]. The high grazing frequency and low resting time of the Boran show its special ability to utilize the available pasture efficiently during grazing time [43].

Another behavioural response to climate stress in ruminants is the reduction in feed intake and utilization in arid regions as adaptive response to regulate internal metabolic heat

production under hot environment [44, 45] and decreasing water excretion by concentrating urine were also alternative strategies to adapt to severe climate stress for livestock in arid and semi-arid of East African rangelands [45, 46]. Increased standing and decreased lying time were also reported to be associated with higher ambient temperatures [47, 48]. Generally, heat stressed animals tend to spend more time standing so that they can reorient themselves in different directions to avoid direct solar radiation and ground radiation. In addition, the standing position also obstructs the conductive heat transfer into the animal body due to the presence of a layer of air adjacent to the skin, and also facilitates the dissipation of body heat load to the surroundings by increasing the amount of skin exposed to air flow or wind.

Boran cattle breed have developed high degree of favourable behavioural adaptation to harsh environment due to genetic basis, human and natural selection for many generations [2]. They have developed ability to withstand periodic shortage of water and feed, ability to walk long distances in search of water and feed and ability to digest low quality feeds from degraded arid rangelands [20]. The Boran cattle are non-selective aggressive grazers and browsers [1]. The Boran herd have a positive social behaviour with good mothering ability which may reduce chance of being attacked by predators which helped them to adapt to free grazing rangelands [41].

**3.1.4. Physiological adaptations.** Some of the physiological mechanisms of adaptations to heat stress are respiration rate, rectal temperature, pulse rate, skin temperature and sweating rate. Increased sweating rate, high respiration rate, vasodilation with increased blood flow to skin surface, reduced metabolic rate, decreased dry matter intake and altered water metabolism are the physiologic responses that have negative impact on the production and reproduction of the cows [49]. All these physiologic responses are substantial and prolonged in *Bos taurus* than in *Bos indicus* [50]. Hence the consequences of exposure to heat stress for production of milk and meat are less pronounced in the tropical cattle [30].

Excessive heat causes decreased food intake and disturbances in protein and energy metabolism, mineral balance, enzymatic reactions, hormones and metabolites secretion in the blood [30]. Metabolic disorders caused by thermal stress led to reduced milk production, growth and reproductive rates and increase the susceptibility of animal diseases causing economic loss [51–53]. Moreover, climate change causes an increase in average temperature and reduced rainfall, putting the sustainability of the livestock production system in risk, especially in countries such as Ethiopia which already has high air temperature averages and grazing systems dependent on the rainy season [6, 7, 54].

The Boran cattle is one of the most productive indigenous cattle breeds in east Africa which is capable of surviving and reproducing under the prevailing harsh climatic, nutritional and management conditions of the region while maintaining good productivity on poor forage and low water availability [1, 37]. This is because they have physiologically adapted and develop lower maintenance requirements than *Bos taurus* cattle [55]. Boran cattle have higher level of lipoprotein-lipase enzyme activity in the subcutaneous fat depot, which enables them to survive drought in the Borana rangelands with drastic recoveries after drought years when pasture condition improves [1, 7, 29]. The Boran cattle are non-selective feeders and browsers which give them the ability utilize shrubs, trees and dry unpalatable grasses that often not consumed by other cattle breeds [6, 34].

**3.1.5. Neuro-endocrine response.** Several studies of various livestock species clearly established higher plasma cortisol level in ruminants during heat stressed conditions [30, 56, 57] reported that the plasma cortisol level was significantly lower in multiple stressors groups (heat, nutrition and walking) as compared with individual (heat stress/nutritional stress) or combined stresses (heat and nutrition stress). Aldosterone is another steroid hormone released from the cortex of the adrenal glands and is involved in the regulation of water and mineral

balance in the body. It is a well-established fact that during heat stress conditions ruminants may undergo severe dehydration, which may result in the activation of renin–angiotensin–aldosterone pathway to restore the water and electrolyte balance [58]. Severe dehydration may lead to increased secretion of antidiuretic hormone (ADH) through activation of renin–angiotensin–aldosterone system. The ADH hormone regulates the blood osmolality by increasing the water absorption in the kidneys, which also assists the excretion of concentrated urine in animals suffering from heat stress [59]. Studies on the responses of neuro-endocrine mechanisms in Boran cattle breed are scarce in Ethiopia and this requires further investigation.

**3.1.6. Blood biochemical response.** Blood biochemical constituents and enzymes are fundamental biomarker for climate stress adaptation in tropical livestock. There are several biochemical and enzymes reported to involve in adaptation of livestock to climate stress. One of these events is an increasing trend of total blood haemoglobin (Hb) with an increase in environmental temperature which could be due severe dehydration [60, 61]. Plasma haptoglobin is also one of the most commonly used acute phase proteins to assess the health and inflammatory response of animals [62, 63] reported a significantly higher production of haptoglobin in the blood plasma of Holstein-Frisian dairy cows exposed to high heat load. In several experiments, significantly increased levels of packed cell volume (PCV) were observed in various livestock species suffering from heat stress ([33]. On the other hand, a decreased concentration of plasma protein [50, 64] and cholesterol [50] were recorded in livestock exposed to elevated ambient temperatures. Further, there are reports which also established an increased concentration of free fatty acid in livestock exposed to heat stress [61].

Antioxidant enzymes such as superoxide dismutase (SOD) and glutathione peroxidase (GPx) are synthesized in the body and provide protection from reactive oxygen species generated during heat stress [65]. These antioxidants scavenge both intracellular and extracellular super oxides and inhibit lipid peroxidation of plasma membrane during the challenges of heat stress [61, 66] reported a significantly higher level of plasma malondialdehyde, SOD and GPx activities in Surti buffaloes during hot humid periods and hot dry periods indicating an increased free radical production during periods of heat stress. In addition to this, plasma antioxidant levels in the hot dry period were significantly higher than in the hot humid period indicating more stressful condition may lead to the elevated synthesis of free radicals [61]. There are also reports establishing significantly higher total antioxidant status (TAS) values in a hot dry season in ruminant animals [61]. All these findings establish the significance of blood biochemical responses to be one of the primary means used by animals to cope with adverse environmental conditions.

**3.1.7. Metabolic responses.** Metabolic adaptation is another important means through which animals tackle challenges of heat stress, essentially by reducing the metabolic heat production [67]. Thyroid hormones play an important role in regulating the thermogenesis and are also identified as an indicator for assessing the thermo-tolerance of the farm animals [68]. Thyroid hormones, namely triiodothyronine ($T_3$) and thyroxine ($T_4$), play a vital role in metabolic adaptation and growth performance of animals [62]. During heat stress, serum and plasma concentrations of $T_3$ and $T_4$ reduce and are likely to be due to the direct effect of heat stress on the hypothalamo-pituitary and thyroid axis to decrease the production of thyrotropin-releasing hormone, which will limit basal metabolism [67]. Reduced concentrations of circulating $T_3$ and $T_4$, were an indicative of an attempt to reduce metabolic rate and thus metabolic heat production in heifers [69].

During periods of high ambient temperatures, some metabolic enzymes increase their activity, the levels of activity of these enzymes in plasma can be informative of how various organs are responding and adapting to heat load and such enzymes play a vital role in the diagnosis of welfare of animals [65]. *Acid phosphatase* (AP) and *alkaline phosphatase* (ALP) are

two major enzymes associated with the metabolic activities in animals. The levels of these enzymes are generally low in heat stressed animals, which could be attributed to a metabolic shift in the animals [65]. Likewise [50], reported a decrease in ALP during summer in ruminants, which they attributed to the dysfunction of the liver during heat stress exposure. *Aspartate aminotransferase* (AST) and *alanine aminotransferase* are two important metabolic enzymes that increase during heat stress exposure in sheep [70] and goats [65]. These authors concluded that such increase in the activity of these enzymes is due to the higher adaptive capability of the animals to cope with heat stress [71].

Another important metabolic regulator is non-esterified fatty acids (NEFA) in plasma and serum [62]. Low NEFA concentrations are mostly reported in heat stressed dairy cows. It is thought that this is an attempt to increase glucose utilization which will result in lower metabolic heat production [72]. However [73], reported an increase in NEFA production of dairy cows during summer compared with winter, which they attributed to an attempt by the animals to maintain energy balance. In summary, at least in livestock, haptoglobin, NEFA, thyroid hormones ($T_3$ and $T_4$) and liver enzymes are considered to be reliable indicators of metabolic adaptation to high heat load [62].

**3.1.8. Cellular and molecular responses.**   The cellular level of adaptation is one of the acute systemic responses to heat stress and it plays a significant role in imparting thermo-tolerance to animals. Gene networks within and across the cells respond to a higher temperature through both intra-and extracellular signals that result in cellular adaptation. Cattles evolved in hot climates had acquired different thermo-tolerant genes when exposed to a higher temperature [2]. Heat stress was found to alter several molecular functions such as DNA synthesis, replication and repair, cellular division and nuclear enzymes and DNA polymerases functions [74, 75]. It also affects a complex array of cellular and molecular responses in livestock [76]. With the development of molecular biotechnologies, new opportunities are available to characterize gene expression and identify key cellular responses to heat stress [25]. For example, there are changes in the expression patterns of certain genes that are fundamental for thermo-tolerance at the cellular level in animals [65]. Such genes having a cellular adaptation function in animals are considered potential biomarkers for understanding stress adaptation mechanisms [30]. The classical *heat shock protein (HSP)* genes, apoptotic genes and other *cytokines* and *toll-like receptors* are considered to be up regulated on exposure to heat stress. Several reports established the role of *HSP70* during heat stress exposure in ruminant livestock and they identified this to be ideal molecular marker for quantifying heat stress response [45, 65]. Apart from this, several other genes such as *Superoxidedismutase* (*SOD)*, *nitric oxide synthase* (NOS), *thyroid hormone receptor* (THR) and *prolactin receptor* (PRLR) genes were found to be associated with thermo-tolerance in ruminant livestock [30].

Furthermore [45], a higher expression of *HSP70 messenger RNA* (mRNA) in the adrenal gland of the multiple stressor groups, which could be attributed to the adaptive mechanism of livestock to counter both the heat stress and nutritional stress. The significantly higher expression of adrenal *HSP70* in the multiple stressed animals as compared with animals subjected only to heat stress could be attributed to additional nutritional stress in the multiple stresses group. The higher *HSP70* expression in the adrenal gland could also be attributed to the hyperactivity of adrenal cortex to synthesize more cortisol as evident from this study [45]. Similarly, the plasma HSP70 and expression pattern of peripheral blood mononuclear cell *HSP70* also showed similar trends of significantly higher value in multiple stressor group animals as compared with control and individual (heat/nutritional) stress groups [45]. Studies have also indicated that HSP70s and their associated cochaperones participate in numerous processes essential to cell survival under stressful conditions. They assist in protein folding and

translocation across membranes, assembly and disassembly of protein complexes, presentation of substrates for degradation and suppression of protein aggregation [76].

## 3.2. Effects of climate change on Boran cattle production

In Arid and semi-arid tropics, climate change has direct negative effect on adaptive capacity [77], growth [72], milk production [78], reproductive [79] and meat production [80] performances of livestock. Further, it can indirectly reduce livestock production by increasing annual variations of the quantity and quality of feed and water resources, reduced dry matter intake and feed utilization, increased thermal stress and sudden disease outbreaks [2, 81]. The adverse impacts of heat stress on these productive functions depend on species and breed differences of livestock and the magnitude of this impact determines the adaptive potential of the animals.

Ethiopian Boran cattle are one of those cattle breeds reported for their special production, reproduction and adaptation traits under such scenario [5], particularly the Borana rangelands of Southern Ethiopia [23, 77]. On other hand, Boran cattle breed is currently challenged by the adverse effects of climate change and other management factors. These are high rate genetic dilution by other small sized zebu cattle through increasing pressure of breed admixture, uncontrolled breeding program and the preference of small sized cattle breed with reduced metabolic requirements as compared large size Borana cattle due to negative effect of climate change on rangeland productivity [6, 77]. There is also conversion of grazing lands to crop lands, bush encroachment and lack of indigenous rangeland management practices [51, 82] hampered rangeland productivity in the region. Lack of organized breed improvement programs and improper selection of gene pool are also critical challenges of Boran cattle breed [7]. However, rate of genetic dilution of Boran cattle by other zebu cattle and the specific reasons why Borana pastoralists decided to admix their pure Boran cattle with local small sized short horn zebu cattle require further investigation.

## 3.3. Production and reproduction characteristics of Borana cattle

**3.3.1. Summary statistics for quantitative traits.** The weighted descriptive statistics and abbreviations for the traits considered in the present study for Boran cattle breed are shown in Tables 1–3.

The number of published articles reported were higher for CI, NSC, DO, AFS, AFC, BW, WW, ADG2, LYD and DMY (Table 2) traits, suggesting their importance for the Boran cattle breeding program. The number of studies as well as the records per articles were comparatively lower for some growth and milk production traits than reproductive traits as these traits are mainly age and sex-specific and difficult to measure periodically. The lowest coefficients of variations were estimated for milk composition traits indicating that the number of studies and records on the specific breed is limited and requires more investigation. In addition, the phenotypic variations for these specific traits are biologically limited [16]. On the other hand,

**Table 1. Mean phenotypic traits of reproduction in Boran cattle breed.**

| Traits | Abb | Unit | Articles | Records | Mean | s. d. | CV (%) |
|---|---|---|---|---|---|---|---|
| Reproduction traits | | | | | | | |
| Calving interval | CI | Days | 16 | 7395 | 421.17 | 126.72 | 30.1 |
| Days open | DO | Days | 14 | 7387 | 162.16 | 35.06 | 21.6 |
| Age at first service | AFS | Months | 13 | 3891 | 34.27 | 6.45 | 18.8 |
| Age at first calving | AFC | Months | 13 | 3638 | 44.23 | 6.47 | 14.6 |
| Number of services per conception | NSC | Number | 17 | 9529 | 1.71 | 0.29 | 17.0 |

**Table 2. Mean phenotypic traits of growth and carcass characteristics in Boran cattle breed.**

| Traits | Abb | Unit | Articles | Records | Mean | s. d. | CV (%) |
|---|---|---|---|---|---|---|---|
| **Growth and carcass traits** | | | | | | | |
| Birth weight | BWT | Kg | 12 | 1762 | 25.43 | 3.49 | 13.7 |
| Weaning weight | WWT | Kg | 10 | 1608 | 116.63 | 57.86 | 49.6 |
| Six-month weight | SWT | Kg | 5 | 296 | 95.5 | 23.33 | 24.4 |
| Yearling weight | YWT | Kg | 8 | 1298 | 209.27 | 52.02 | 24.9 |
| 18 months weight | EWT | Kg | 6 | 491 | 212.63 | 56.01 | 26.3 |
| Adult weight | AWT | Kg | 8 | 463 | 356.85 | 91.51 | 25.6 |
| Average daily gain from birth to weaning | ADG1 | G | 10 | 865 | 402.37 | 120.26 | 29.9 |
| Average daily gain from weaning to adult age | ADG2 | G | 7 | 802 | 243.66 | 84.93 | 34.9 |

the higher coefficient of variations was observed for WWT (49.6%), Omega-FA (36.8%), ADG (34.9%) and CI (30.1%), showing that there is larger phenotypic variation in these traits than in others (1–3 Tables).

**3.3.2. Reproduction and production performances of Boran cattle and other cattle breeds.** *3.3.2.1. Reproductive characteristics.* Comparisons of the reproductive performance of Ethiopian Boran with other indigenous cattle breeds indicated that Boran cattle have better reproductive performance (Table 4). The breed has shorter female age at first matting (FAFM), age at first calving (AFC), calving interval (CI), days open (DO) while longer reproductive life time for male (RLTM) and reproductive life time for female (RLTF) as well as comparable male age of fertility for mating (MAFM). These traits vary from low to high in heritability and are more challenging to record [83].

In Ethiopia, breeding mainly relies on natural service and, therefore, acceptable bull fertility is also critical [5]. Factors that determine a bull's fertility and performance include plane of nutrition [5], structural soundness, capability of the reproductive organs, quality of the semen, libido level and servicing capacity [88]. Under pastoralist management conditions, age at first calving of Borana cattle is about four years of age [89]. At Abernossa ranch in Ethiopia, weight

**Table 3. Mean phenotypic traits of milk yield and composition in Boran cattle breed.**

| Traits | Abb | Unit | Articles | Records | Mean | s. d. | CV (%) |
|---|---|---|---|---|---|---|---|
| **Milk yield and composition traits** | | | | | | | |
| Lactation yield | LYD | Kg | 10 | 3256 | 596.3 | 165.12 | 27.7 |
| 305 days yield | 305YD | Kg | 7 | 2751 | 506 | 91.72 | 18.1 |
| Lactation length | LL | Days | 9 | 2901 | 230.76 | 40.66 | 17.6 |
| Daily milk yield | DYD | Kg | 10 | 3627 | 2.02 | 0.48 | 23.9 |
| Total lactation yield | TLYD | Kg | 8 | 2226 | 2440.51 | 672.36 | 27.6 |
| Milk fat | MFC | % | 7 | 4255 | 5.01 | 1.23 | 24.7 |
| Milk protein | MPC | % | 7 | 1751 | 3.63 | 0.37 | 10.1 |
| Lactose content | MLC | % | 7 | 459 | 4.65 | 0.60 | 12.9 |
| Solid not-fat | TSNF | % | 7 | 1714 | 8.93 | 0.27 | 3.02 |
| Total solid | MTSC | % | 7 | 2531 | 14.78 | 1.07 | 7.3 |
| Short chain FA | TSFA | % | 5 | 125 | 64.5 | 4.33 | 6.7 |
| Medium chain FA | TMCFA | % | 5 | 125 | 33.2 | 3.81 | 11.5 |
| Polyunsaturated FA | TPUFA | % | 5 | 125 | 2.79 | 0.6 | 21.5 |
| Omega-3 FA | W-3FA | % | 5 | 125 | 0.38 | 0.14 | 36.8 |
| Omega-6 FA | W-6FA | % | 5 | 125 | 2.41 | 0.47 | 19.5 |

**Table 4. Average reproduction and production performances of Boran cattle as compared to other Ethiopian cattle breeds.**

| Breed | Borana | Horro | Begailt | Arsi | Fogera | Sheko |
|---|---|---|---|---|---|---|
| **Reproduction traits** | | | | | | |
| MAFM | 47.4 | 46.9 | 42.3 | 36.3 | 45.4 | - |
| FAFM | 37.6 | 54.8 | 39.3 | 41.8 | 42.2 | - |
| AFC | 48.6 | 50 | 53.1 | 49.1 | 55.1 | 42.2 |
| CI | 15.3 | 17 | 18.2 | 14.5 | 21.2 | 16.5 |
| RLTM (yr) | 9.6 | 7.2 | 6.8 | 7.4 | 6.8 | 6.5 |
| DO (days) | 11.2 | 13.6 | 9.6 | 12.1 | 11.3 | 14.7 |
| **Milk production and growth traits** | | | | | | |
| DMY (kg) | 3.2 | 1.65 | 2.44 | 1.82 | 2.93 | 2.79 |
| LMY (kg) | 596.3 | 512.9 | 539.1 | 809 | 777 | 774 |
| LL(Months) | 8.7 | 8.64 | 6.0 | 9.3 | 16.9 | 10.1 |
| BW (kg) | 23.1 | 19.9 | 22.6 | 17.3 | 21.9 | 16.1 |
| MW (kg) | 394.2 | 250 | 294 | 264 | 384 | 275 |

Source: [5, 78, 84–87]

and age at puberty in heifers were found to be about 155 kg and 22 months, respectively. Calving rate under a single-sire mating system was also improved to above 80% [5], compared to about 45% under pastoral management conditions [89]. These results would therefore indicate the improvement that could be achieved through proper selection scheme and better management. On the other hand, Ethiopian Boran had longer calving interval, lower breeding efficiency, delayed age at first service and age at first calving and longer days' open compared with their Friesian crosses. These differences in reproduction performance between Ethiopian Boran and the crosses are comparable with those obtained for local cattle and their crosses in Ethiopia [84, 90]. Thus, the more advanced age at first calving obtained in the Boran compared with their exotic crosses indicated the potential that could be exploited by merely improving management. However, there was no significant difference in number of services per conception among the genetic groups [5]. This could due the fact that nutrition [84] and inseminator effects [29] are more important contributors to the variation in number of services per conception than genotype.

*3.3.2.2. Growth and carcass characteristics.* Table 4 indicates comparative summary of birth weight (BW) and mature weight (MW) of six Ethiopian cattle breeds. Boran cattle had better performance in both growth traits (Table 4). Additionally, performance of Boran has been substantially improved through improvement in management and selection under intensive system. For example, the improved Boran had birth weight of 30kg in Kenya [4] and weaning weight of 158 kg at Abernossa ranch in Ethiopia [84]. This variation indicates the potential that can be exploited by within breed selection and improvement in management. Growth performance of Boran and their crosses as beef animal has been evaluated in different parts of the world. For example, the value of the Boran for beef production in the tropics is clearly shown in crossbreeding studies in Kenya (www.borankenya.org). In this experiment the F1 Angus/Boran steer weighed 426 kg when sold at 13 months with a daily gain of 1.36 kg. In Australia (www.boransaustralia.com), it is generally believed that using Boran bulls for crossbreeding is the quickest way of improving the commercial potential of beef herds, because one of the most important attributes of the Boran is its ability to transmit hybrid vigour to the traditional beef breeds of Australia (including Brahman and its' crosses). The higher growth performance observed for the crossbred calves in comparison with the Ethiopian Boran cattle could be due

to the effects of breed and heterosis on the growth performance of crossbred cattle [5]. The average adult live weight differs depending on the status of improvement, level of management and production system. For example, mature bulls of the Improved Boran in Kenya were reported to weigh 550–850 kg and those for cows were 400–550 kg [4]. The birth weight under range condition of Ethiopia is indicated to be 18 kg, while it is nearly 25 kg in research stations of Ethiopia and in the commercial ranches of Kenya [2]. Due to this good beef performance, it is indicated that Boran could be used to improve small East African Zebu [5].

The Boran had higher carcass weight when compared with Angoni and Barotse breeds of the same age group, while Boran and Angoni were comparable with regard to dressing percentage [77]. Carcass characteristics differ between breeds and are influenced by the plane of nutrition and production system [87]. Selection for these traits is greatly influenced by the market demand. In the Ethiopian context, export markets demand lean meat whereas when the target is local market, fattened cattle are required [5, 87]. Therefore, the breeding, feeding and other management conditions should be designed in such a way that the requirements of the specific market are met.

It is established that Boran produces high quality beef by utilizing low quality forage. This is substantiated by data from the FAO/UNDP feedlot trial at Lanet in Kenya, where 7625 Boran and crossbreds were fed between 1968–73 (www.borankenya.org). Similarly, the performance of improved Boran as a beef breed was reported higher in South Africa, USA and Australia [4]. In addition, Boran cattle are less affected by mild drought shocks and have fast recovery rate [24]. In Ethiopia, literature report on beef attributes of Ethiopian Boran is scanty. Some of the literatures on growth performance of Ethiopian Boran cattle presented in Table 3 are based on data from crossbreeding studies where Ethiopian Boran was used for dairy production. There is therefore, a need to investigate the beef qualities of the Ethiopian Boran for future uses.

*3.3.2.3. Milk yield and quality characteristics.* Milk production performance of Ethiopian Boran compares fairly well with other indigenous Ethiopian breeds (Table 4). Boran cattle had better daily milk yield (DMY), lactation milk yield (LMY) and comparable lactation length (LL) among Ethiopian cattle breed under grazing systems. However, the improved Boran in Kenya had much higher performance than the unimproved Ethiopian Boran. In Kenya, improved Boran cows could produce 1130 kg milk over a 36-week period during with calves suckled three times a day producing butter fat percentage of 5.7, 5.8, 5.9 and 6.1, respectively [4]. But under more favourable production environment, Boran cows produced up to 1657kg of milk per 252-day lactation [2]. On the other hand, converting calf growth up to 7 months in to milk intake and adding recorded values of milk off-take for human consumption; lactation milk yield of Boran cows was calculated to be 843 kg under the pastoralist system in Ethiopia [90]. Moreover, crossbreeding of Ethiopian Boran with Holstein Friesian resulted in improved milk production [5]. For example, 50% Holstein Friesian crosses had a fourfold increase over the Ethiopian Boran breed in terms of lactation milk yield, 305-days milk yield, daily milk yield and life time milk yield; they were also milked for 97 more days than Ethiopian Boran.

In general, meta-analysis for performances of Boran cattle compared with other Zebu cattle breeds as beef and/or dairy cattle are described by four categories of traits namely, reproduction, growth, milk production, and carcass yield and quality traits (1–3 Tables). Although their production potential is less compared with exotic and cross breeds, the level of production of Boran cattle is relatively stable during harsh conditions where high producing animals are at risk [1]. Furthermore, during periods of extreme heat stress, water scarcity and reduced pasture availability, they maintain their reproductive potential due to their smaller body size whereas the larger exotic animals may face reproductive impairments which could be attributed to their higher maintenance energy requirements [5, 24].

*3.3.2.4. Heritability of production and reproduction traits in Boran cattle*. Direct and maternal–heritability estimates of reproductive and production traits with respective 95% confidence intervals and the $I^2$ index to test the significance of heterogeneity among studies, for each trait, for Boran cattle are shown in Table 4. Direct heritability estimates for reproduction traits ranged from lower to medium magnitude. Lower heritability estimates were observed for DO (0.040) and CR (0.042), for which the impudence of environmental factors is more evident. Medium heritability was estimated for longevity (0.08), SPC (0.081), CI (0.03) and AFC (0.215), indicating relatively better response to selection for these traits as compared to the other indicated reproductive traits. Usually, indigenous cattle in Ethiopia are raised on extensive production systems, which are often characterized by high temperatures and periods of feed and water scarcity that might affect the reproductive performances of animals [5].

These phenomena are quite severe for poorly adapted exotic and crossbreds [91], but for Boran cattle breed which had better adaption to tropical environment, the observed heritability estimates were from lower to medium (Table 5), implicating selection response for AFC should be faster than other traits for Boran cattle. Similar trend was observed for reproduction traits of Nellore cattle in Brazil [16].

Direct heritability estimates for the growth traits ranged from medium in BWT (0.25), WW (0.243), and YW (0.265) to higher in ADG (0.46) showing that relatively higher response to selection for ADG trait than other growth traits in Boran cattle breed (Table 5). It shows that

**Table 5. Number of articles to estimate direct and maternal heritability ($N_d$ and $N_{mat}$ respectively), direct and maternal heritability ($h^2$ and $h^2$ mat respectively) estimated from meta-analysis using random-effects model, 95% confidence interval for heritability (95% CI and 95%CI $_{mat}$ respectively) and the $I^2$ index to test heterogeneity in each trait ($I^2$ and $I^2$ mat respectively).**

| Trait | $N_d$ | $h^2$ | 95% CI | | $I^2$ (%) | Repeatability (%) | $N_{mat}$ | $h^2_{mat}$ | 95% CI $_{mat}$ | | $I^2_{mat}$ (%) |
|---|---|---|---|---|---|---|---|---|---|---|---|
| | | | LLQ | ULQ | | | | | LLQ | ULQ | |
| **Reproduction traits** | | | | | | | | | | | |
| AFC | 6 | 0.22 | 0.20 | 0.23 | 99.1 | 51.2 | | | | | |
| CI | 5 | 0.08 | 0.01 | 0.16 | 92.8 | 62.5 | | | | | |
| SPC | 7 | 0.08 | 0.04 | 0.12 | 87.1 | 54.7 | | | | | |
| DO | 6 | 0.04 | 0.03 | 0.05 | 72.4 | 48.2 | | | | | |
| CR | 8 | 0.04 | 0.02 | 0.06 | 91.3 | 71.7 | | | | | |
| Longevity | 4 | 0.08 | 0.05 | 0.11 | 83.2 | 88.5 | | | | | |
| *Growth traits* | | | | | | | | | | | |
| ADG | 7 | 0.46 | 0.34 | 0.58 | 72.1 | | 4 | 0.2 | 0.06 | 0.34 | 69.6 |
| BWT | 9 | 0.25 | 0.14 | 0.36 | 80.7 | | 6 | 0.09 | 0.04 | 0.15 | 82.1 |
| WW | 6 | 0.24 | 0.13 | 0.35 | 92.1 | | 4 | 0.14 | 0.07 | 0.21 | 90.1 |
| YW | 8 | 0.27 | 0.07 | 0.46 | 87.2 | | 7 | 0.20 | 0.13 | 0.26 | 92.4 |
| *Milk yield traits* | | | | | | | | | | | |
| DMY | 8 | 0.13 | 0.09 | 0.17 | 67.4 | 32.7 | 5 | 0.34 | 0.15 | 0.53 | 56.9 |
| LYD | 7 | 0.19 | 0.13 | 0.26 | 45.2 | 38 | 6 | 0.06 | 0.03 | 0.10 | 49.2 |
| LL | 5 | 0.12 | 0.07 | 0.16 | 67.4 | 37.7 | 4 | 0.2 | 0.14 | 0.26 | 68.3 |
| LYD | 6 | 0.20 | 0.05 | 0.07 | 50.4 | 26 | | | | | |
| 305YD | 5 | 0.18 | 0.04 | 0.07 | 55.2 | 23 | | | | | |
| LL | 4 | 0.26 | 0.10 | 0.30 | 60.7 | 46 | | | | | |
| *Milk composition traits* | | | | | | | | | | | |
| Fat% | 3 | 0.49 | 0.43 | 0.55 | 91.2 | 0.98 | | | | | |
| protein% | 3 | 0.26 | 0.32 | 0.34 | 62.1 | 0.59 | | | | | |
| SNF% | 3 | 0.46 | 0.44 | 0.50 | 72.4 | 0.93 | | | | | |
| TS% | 3 | 0.45 | 0.52 | 0.57 | 91.2 | 0.99 | | | | | |

ADG, birth weight and mature weight have the highest heritability among the production traits which had good correlation with FCR explaining that accelerated genetic progress could be obtained for these traits through selections in Boran cattle [23].

Similarly, direct heritability estimates for milk production and composition traits ranged from medium to higher magnitude, indicating better response for selection for milk composition traits compared to milk yield traits in Boran cattle. The heritability estimates for growth for Nellore cattle 16] and other Zebu cattle breeds [92] corroborates the current study for Boran cattle.

Heterogeneity among studies were moderate in milk yield traits by $I^2$ index (50 to 75%) and higher in reproduction, growth and milk composition traits with $I^2$ index (>75%). These results reinforced the importance of accounting this heterogeneity in selecting random effect model in the meta-analysis reviews of the studies (Table 5).

The weight and milk production performances of Boran calves are also affected by the maternal genetic effect from dam. In this study, maternal-heritability estimates for growth traits had a low to medium magnitude (ranging from 0.089 to 0.20), indicating that the genetic progress for these maternal effects is slow in this breed. Similar trend was observed for milk production traits. It is well known that the initial phases of growth traits are influenced by the maternal effect and ignoring these effects may lead to bias in many situations [95]. In general, higher heritability estimates were reported in the studies for milk composition traits followed by growth traits and milk yield traits but lower heritability estimates were reported for reproduction traits in Boran cattle except trait for AFC. Since meta-analysis brings together published parameter estimates provided by studies based on populations at different stages of selection, with different sample sizes and considering distinct effects in the model, it is expected that the true parameter may vary from study to study. In meta-analysis, the estimated confidence intervals of the studied traits were narrower than those obtained from individual published articles. In Boran cattle, data on heritability estimates for carcass traits were limited for this meta-analysis which requires more studies in the future.

On the other hand, heritability for average daily gain and feed conversion ratio were estimated to be medium for other African cattle breeds (Table 6). The summary of ranges of

**Table 6. Summary of ranges of heritability values for reproduction and production traits of Tropical cattle.**

| Reproductive traits | $h^2$ | References |
|---|---|---|
| Age at first calving | 0.04–0.31 | [91, 93] |
| Calving date | 0.02–0.09 | [93] |
| Calving success | 0.03–0.27 | [93] |
| Calving rate | 0.04 | [91] |
| Calving interval | 0.02–0.13 | [91, 93] |
| Days open | 0.04 | [84] |
| Longevity | 0.08 | [94] |
| **Production traits** | | |
| Birth weight (direct) | 0.21–0.4 | [95, 96] |
| Birth weight(maternal) | 0.05–0.14 | [95, 96] |
| Weaning weight(direct) | 0.12–0.29 | [95, 97] |
| Weaning weight(maternal) | 0.11–0.21 | [95, 97] |
| Yearling weight | 0.13–0.26 | [95, 96] |
| Final weight | 0.13–0.42 | [93, 95] |
| Mature weight | 0.24–0.41 | [96, 98] |
| Average daily gain (ADG) | 0.38 | [99] |
| Feed conversion ratio (FCR) | 0.23–0.41 | [100] |

heritability for tropical cattle breeds indicate that they have lower heritability for reproductive traits except for AFC and lower to medium values for production traits. Similar trends were observed in the current meta-analysis for both reproduction and production traits of Boran cattle.

*3.3.3.5. Genetic correlations between reproduction and production traits in Boran cattle.* Genetic-correlation estimates with the respective 95% confidence interval and the $I^2$ index to quantify the degree of heterogeneity among the studies were shown in Table 7 for each trait pairs.

Except correlation involving AFCXCI, CDXCI and DOXPR which assumed negative values, all other correlation estimates were positive. Higher correlation estimates were observed between WWXYW, WWXMW, PXSN, PXTS, LLXYD and SNFXTS trait pairs, while medium correlation estimates for WXWW, BWXMW, and FXP and lower correlation estimates for 305YDXLL, 305YDXLD, 305YDXDMY, LYDXDMY, FXSNF and BWXYW for Boran cattle breed.

Similar to heritability estimates, the estimated confidence interval of genetic correlations was narrower than reported in individual published studies. All correlation estimates presented higher heterogeneity among studies by the $I^2$ index ($I^2 > 75\%$) which justify the use of random-effects model. In general, a small number of articles reporting genetic correlation estimates were found for purebred Boran cattle. This highlighted the requirement for studies investigating these estimates, which, in general, also presented high standard errors in relation to heritability estimates (Table 7). Estimated genetic correlations among some reproduction and production traits in these studies are in line with summary of genetic correlations of fertility and production traits for tropical zebu cattle in East Africa (Table 8). Except for correlation between AFC and CD, the other correlation coefficient between pairs of reproduction traits

**Table 7. Number of articles ($N_d$), genetic correlation among traits($g_r$) estimated from meta-analysis using random model, 95% confidence interval (95% CI), and the values of $I^2$ index($I^2$) in each trait.**

| Traits | $N_d$ | $r_g$ | 95%CI | | $I^2$ (%) |
|---|---|---|---|---|---|
| | | | LLQ | ULQ | |
| AFCXCD | 4 | 0.090 | 0.086 | 0.094 | 88.4 |
| AFCXCI | 4 | -0.030 | -0.035 | -0.024 | 97.1 |
| CDXCI | 5 | 0.010 | 0.008 | 0.012 | 84.9 |
| DOXPR | 4 | -0.99 | -0.991 | -0.988 | 95.3 |
| BWXWW | 4 | 0.560 | 0.285 | 0.834 | 88.3 |
| BWXYW | 3 | 0.305 | 0.156 | 0.456 | 91.6 |
| BWXMW | 5 | 0.587 | 0.373 | 0.801 | 89.2 |
| WWXYW | 4 | 0.600 | 0.426 | 0.774 | 82.5 |
| WWXMW | 5 | 0.894 | 0.733 | 1.056 | 99.9 |
| LLXLYD | 4 | 0.885 | 0.679 | 1.09 | 99.7 |
| 305YDXLL | 5 | 0.410 | 0.155 | 0.665 | 87.5 |
| 305YDXLYD | 3 | 0.340 | 0.242 | 0.438 | 83.8 |
| 305YDXDMY | 3 | 0.297 | 0.120 | 0.474 | 99.9 |
| LYDXDMY | 3 | 0.524 | 0.329 | 0.720 | 93.7 |
| FXP | 2 | 0.510 | 0.275 | 0.745 | 99.7 |
| FXSNF | 2 | 0.160 | 0.062 | 0.258 | 82.7 |
| FXTS | 2 | 0.920 | 0.626 | 1.214 | 86.4 |
| PXSNF | 2 | 0.710 | 0.612 | 0.808 | 93.1 |
| PXTS | 2 | 0.820 | 0.644 | 0.996 | 99.1 |
| SNFXTS | 2 | 0.780 | 0.662 | 0.898 | 92.7 |

**Table 8. Genetic correlations (rg) for specific fertility and production traits in tropical cattle breeds.**

| Traits | $h^2$ | References |
|---|---|---|
| **Reproductive traits** | | |
| Age at first calving and calving date | 0.09–0.88 | [93, 94] |
| Age at first calving and calving interval | -0.03–0.44 | [93, 94] |
| Calving date and calving interval | 0.01–0.75 | [93, 94] |
| Calving success and calving date | -0.95 | [94] |
| Days to calve and pregnancy rate | -0.99 | [93] |
| **Production traits** | | |
| Birth and weaning weight | 0.45–0.78 | [93, 96] |
| Birth and yearling weight | 0.28–0.57 | [96, 101] |
| Birth and final weight | 0.45–0.6 | [96, 102] |
| Birth and mature weight | 0.63 | [96] |
| Weaning and yearling weight | 0.86 | [96] |
| Weaning and final weight | 0.71–0.99 | [96, 102] |
| Weaning and mature weight | 0.94 | [96] |
| Yearling and final weight | 0.85 | [96] |
| Yearling and mature weight | 0.43 | [96] |
| Final and mature weight | 0.75 | [96] |

were negative which may explain improvement for one trait may negatively affect the performance of the other traits [93].

Although genetic correlations estimated in the present study were in favour of the common objectives of genetic improvement programs of Boran cattle in relation to climate change adaptation, it is important to re-estimate the genetic parameters in regular intervals, especially if selection of different traits is made simultaneously [5].

## Summary

The Boran cattle is a hardy local cattle breed originated from Borana range lands in the southern Ethiopia and kept for meat and milk production. The breed has special merits of having the ability to survive, produce and reproduce under high ambient temperature, utilize low quality forage resources, resist water shortage or long watering intervals and tick infestations. The review also revealed that Boran cattle employ various adaptation responses (morphological, physiological, biochemical, metabolic, cellular and molecular responses) to cope with harsh environmental conditions including climate change, rangeland degradation, seasonal feed and water shortages and high incidences of tick infestations.

The meta-analysis using a random-effects model allowed provision of pooled estimates of heritability and genetic correlations for reproduction and production traits, which could be used to solve genetic prediction equations under a population level in purebred Boran cattle. In addition, heritability and genetic-correlation estimates found in the present study suggest that there is high genetic variability for most traits in Boran cattle, and that genetic progress is possible for all studied traits in this breed. However, the Ethiopian Boran cattle breed is facing several challenges such as recurrent droughts, pasture deterioration, lack of systematic selection and breeding programs and threat from genetic dilution due to the admixture of other breeds. Thus, we recommend systematic selection for improving the reproductive and production performances without compromising the adaptation traits of the breed coupled with improved management of rangelands.

## Supporting information

**S1 Checklist.**
(DOC)

**S1 Data.**
(XLSX)

**S2 Data.**
(CSV)

**S3 Data.**
(XLSX)

## Author Contributions

**Conceptualization:** Sintayehu Yigrem, Simret Betsha, Adugna Tolera.

**Data curation:** Merga Bayssa.

**Formal analysis:** Merga Bayssa.

**Funding acquisition:** Sintayehu Yigrem, Simret Betsha, Adugna Tolera.

**Investigation:** Merga Bayssa.

**Methodology:** Merga Bayssa.

**Project administration:** Sintayehu Yigrem.

**Resources:** Sintayehu Yigrem, Simret Betsha, Adugna Tolera.

**Software:** Merga Bayssa.

**Supervision:** Sintayehu Yigrem, Simret Betsha, Adugna Tolera.

**Validation:** Merga Bayssa.

**Visualization:** Merga Bayssa.

**Writing – original draft:** Merga Bayssa.

**Writing – review & editing:** Sintayehu Yigrem, Simret Betsha, Adugna Tolera.

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
