## [Decision Letter · Decision Letter 0]

23 Mar 2021

PONE-D-20-39503

Production, Reproduction and Adaptation characteristics of Boran cattle breed under changing climate: A systematic review and meta-analysis

PLOS ONE

Dear Dr. Bayssa,

Thank you for submitting your manuscript to PLOS ONE. After careful consideration, we feel that it has merit but does not fully meet PLOS ONE’s publication criteria as it currently stands. Therefore, we invite you to submit a revised version of the manuscript that addresses the points raised during the review process.

ACADEMIC EDITOR:

Despite the fact that your study delivers an interesting data both reviewers have raised several issues which need your consideration in revising the manuscript.  

We look forward to receiving your revised manuscript.

Kind regards,

Dawit Tesfaye

Academic Editor

PLOS ONE

Journal Requirements:

'This study was conducted in the framework of the German-Ethiopian SDG Graduate School

Climate Change Effects on Food Security” (CLIFOOD) between the Food Security Centre,

University of Hohenheim (Germany) and the Hawassa University (Ethiopia). The German

Academic Exchange Service (DAAD) and the Federal Ministry for Economic Cooperation and

Development (BMZ) financed the research under CLIFOOD.'

No, The funders had no role in study design, data collection and analysis, decision to publish, or preparation of the manuscript.

Additional Editor Comments (if provided):

Reviewers' comments:

Reviewer's Responses to Questions

**Comments to the Author**

1. Is the manuscript technically sound, and do the data support the conclusions?

Reviewer #1: Yes

Reviewer #2: Partly

2. Has the statistical analysis been performed appropriately and rigorously? 

Reviewer #1: Yes

Reviewer #2: I Don't Know

3. Have the authors made all data underlying the findings in their manuscript fully available?

Reviewer #1: Yes

Reviewer #2: Yes

4. Is the manuscript presented in an intelligible fashion and written in standard English?

Reviewer #1: Yes

Reviewer #2: No

5. Review Comments to the Author

Reviewer #1: This is an interesting, well-written study that worth publishing in Plos One journal. However, few sections need attention before publication. My comments and concerns are appended below as well as in the pdf of the article.

Abstract:

The abstract is a little bit wordy. Please provide a compact introduction with proper justification and add the results in the result section. It may appear that the conclusion part contains some of the results, particularly, the first sentence. Necessary actions need to be taken in the abstract part to make it more attractive.

Introduction:

This part is well written and gives the necessary information.

Methodology:

Search and Selection of studies: Why did the authors use a wide range of databases. Wouldn’t it be better to use only the recognized databases such as Scopus and web of science?

Did the authors consider the geographical distribution? Cattles in Kenya may not face similar challenges that are faced by cattle in Ethiopia. The authors need to mention the geographical consideration in the selecting criteria as one of the major goals of the article is to highlight the adaptation strategies of Boran cattle under climate change.

Result and Discussion:

Well written.

Summary:

Self-contradictory statements in the summary part. If the Boran cattle successfully adopted several strategies against climate change (mentioned in 682-684 lines), how could it be under threat due to climatic change (693-694 lines)? This section needs clarification.

General comment:

Careful copy editing is required, particularly, spacing. I have marked a few in the pdf file. The authors should carefully check throughout the manuscript.

Reviewer #2: In this article, the authors aimed to compile the main production, reproduction and adaptation traits of Boran cattle based on a combination of systematic review and meta-analysis of peer-reviewed and published articles on this subject. The research question was “what are main production, reproduction and adaptation characteristics of Ethiopian Boran cattle breed as compared to other zebu cattle in the country?”

The authors followed the PRISMA guidelines for the implementation of the systematic and meta-reviews. They reported the inclusion and exclusion criteria for the obtained publications. However, some major revisions are necessary.

- Did the authors register their protocol and research plan in any of the organizational databases for systematic reviews?

- Quality control: In this step (Line 156), the authors reported that they controlled the quality of quantitative traits based on the relative standard error calculations. What about the adaptive traits? The quality control step is not only for the traits but it should be for each included study. A clear quality assessment criteria that lead to a specific quality score (or classified as high, moderate, or low) for each study should be explained. Several Quality Assessment Tools are available online that can help the authors to quality control the selected studies and decide to include or exclude these studies based on the quality score. As written in line209: “72 articles were subjected to full-text assessment for validity”. This full-text assessment method and criteria are not explained clearly in the article.

In addition, the authors should mention how many investigators were assessed and selected the studies and the cutoff score of which the studies were selected based on it. In the case of 2 investigators were assessed the studies, how were the disagreements resolved?

- Fig.1: Please add in the boxes of the excluded records the reasons for which these records have been excluded and the numbers of excluded records per each reason.

- Table 1: This table needs to be removed or replaced. The information in this table is already mentioned in the text (Line 221-223). Instead, a table with the characteristics of the included studies and their main outcomes concerning the selected traits will be more informative.

- Line 239: This subsection should be under the index number 3.2.1.1, not 3.2.2.1 and the following subsections should be also adjusted.

- In most of the Adaptation characteristics of Boran cattle breed subsections, the authors didn’t present any specific information about the Boran cattle. Instead, they presented general information about cattle. This was clearly shown in the subsections: Physiological adaptations, Neuro-endocrine response, Blood biochemical response, Metabolic responses, Cellular and molecular responses. For example: as mentioned in table 1, 11 studies related to Neuro-endocrine response were included. However, in the result section (Line 336), no information or data related to Boran cattle was discussed. The same was for Blood biochemical response and the subsequent parameters.

- In the same section of Adaptation characteristics of Boran cattle breed, most of the discussed studies that related to Boran cattle were either previously published review articles (which mentioned in the exclusion criteria) or other non-research article sources. For example, in the Physiological adaptations subsection (Line 326), references 1,7,6,33 are review articles, and Ref. 29 is not a research article. The same was in the other traits as in References 20,32, 34.

- Based on the above comments, the first part of this systematic review (Adaptation characteristics) is not adequate to support the objective of the study and should be improved.

- The use of trait names and abbreviations in the different tables and Fig.2 should be consistent. Some tables representing the full name of traits and some others with only undefined abbreviations as in Fig 2.

- Line 481: The first paragraph of the “3.3.2. Quality control” is a kind of repetition of information that is already mentioned in the subsection “2.4. Quality control”.

- Typing errors need to be revised in the whole manuscript. For example:

Line 42 and 698: “adapataion”

Line 43: “pottential”

Line 77: “comprehenssive”

Line 210: “exraction”

Line 211: “fullfill”

Line 229: “varaibilties”

6. PLOS authors have the option to publish the peer review history of their article (what does this mean?). If published, this will include your full peer review and any attached files.

Reviewer #1: **Yes: **Md Mahmodul Hasan Sohel

Reviewer #2: No

---

## [Author Response · Author response to Decision Letter 0]

20 Apr 2021

Response to comments of Reviewer #1

The authors corrected all the comments given in the pdf format of the papers as well as in the separate sheet and incorporated in the improved version of the paper.

Abstract

The introduction subsections are modified as indicated in track changes in manuscripts and included in final copy. Lines 7-18 are replaced/modified to make the introduction attractive short and precise.

Methodology

We have included search databases from Scopus and web of science which are recognized by the international community but unfortunately published papers on specific study topics on these sites were quite few for Boran cattle of Ethiopia. We decided to include more searches from AGORA, Google Scholar, Google web, Science Direct, CAB direct, African Journals online (AJOL) and lists of references of articles from peer reviewed publications. Yet by having wide arrays of these search categories, the available published papers on Boran cattle breed of Ethiopia were limited. 

We have taken comments on the geographical distribution in account for adaptation strategies of Boran cattle in the selection criteria for which we have utilized adaptation characteristics of Boran cattle or Zebu cattle breeds of East Africa (north Kenya) which is immediately adjacent to Boran rangelands of Southern Ethiopia. Both pastoralists are of the same community living in two countries rearing fairly similar breeds of cattle under same pastoral production systems This is the reason why we have used limited literature from Kenyan Boran. 

Summary 

The pure Ethiopian Boran cattle reared by the pastoral community in the Borana rangelands were under threat of genetic dilution due to intensive admixture with either local zebu cattle breeds or small sized Boran subtypes. These might be due to either limited capacity of the Borana community to replace pure Boran cattle breed just after climate related severe drought shocks such as depletion in feed and water, rangeland degradation and high price of large frame size Boran cattle with their high maintenance requirement forced the pastoralists to use small sized Boran ecotypes or other local zebu cattle for restocking could potentially compromise the pure Boran cattle population. We understand that the above reasons may require further investigation. 

General comments

Corrections were for all general comments given line by line with careful typographic corrections and spacing errors.

Responses to comments of Reviewer #2

Did the authors register their protocol and research plan in any of the organizational databases for systematic reviews?

The authors did not register the protocol and the search plan utilized in this paper, rather the authors adopted the already established PRISMA statement for reporting systematic reviews and meta-analysis of studies published on PLoS MEDICINE with some modification as mixed method systematic reviews and meta-analysis as indicated in guideline for preparation of a compressive literature reviews (wood et al.,2016 on Systematic reviews, Biomed central and Lee et al., 2015 on international journal of behavioural, Nutrition and Physical studies). We incorporated the detail of inclusion criteria, search strategy and selection and data extraction in the improved version of this paper both as track changes and final copy. 

Quality assessment tools and Quality control

We have admitted that we did not indicated the quality assessment for the qualitative data collected for systematic reviews. We followed the mixed method appraisal tool (MMAT) as described by Wood et al., (2016). After reviewing the studies, extracting study data and completing summary tables the methodological quality was assessed using the Mixed-Methods Assessment Tool by two reviewers. Reviewers met then to discuss discrepancies as well as the overall strengths and limitations of the studies. Discrepancies were overcome through consensus. Within and across study analysis and synthesis of the findings sections was undertaken using thematic synthesis. Studies were included in the final review provided their score was 50 % or greater, according to the scoring scheme of Pluye et al., 2009. The Grading of Recommendations, Assessment, Development and Evaluation (GRADE) guidelines were used to present a summary of findings and to qualify the strength of individual findings for the quantitative and qualitative syntheses separately, using GRADE profiler and the Confidence in the Evidence from Reviews of Qualitative Evidence (CERQual) tool, respectively. The items were graded as high, moderate, low and very low categories. In general, those evidences with high and moderate classes were included for this particular review.

Response to comments on Fig 1, Figure 2 and Table 1

In Fig1 the inclusion and exclusion criteria of the studies were well taken and elaborations were also given accordingly in separate sheet as per guidelines of the journal. Similarly, Figure 2 abbreviation were included in the box plot.

Table 1 is removed and replaced by information given in text Line 221-223 as per the comments given by the reviewer and the contents were also corrected in the same way. 

Line 239 subsection 3.2.2.1. were moved to index 3.2.1.1 and the subsections are adjusted. 

Response to comments on Adaptation characteristics

In the search of the papers for these reviews, we faced limitation in publications on the specific quantitative adaptation characteristics of the Boran cattle in Ethiopia during our database search. Thus, we refined our search inclusion to published papers, unpublished literatures and reports for synthesis of the adaption characteristics of the Boran cattle. When the information were available specific adaptation characters of Boran cattle were synthesized but for some adaptation characteristics indicated by the reviewers we decided to utilize the information available on the East African zebu cattle breed which have general similarity with Boran cattle in some adaptationcharacteristics. We made major revision to exclude literatures obtained from non-zebu cattlebreeds and those obtained from cattle of non-tropicalorigins. In general, very limited quantitative information were available on adaptation traits of Boran cattle in relation climate change and this will be future research objectives which was proposed by the authors in the final revision.

Physiological adaptations subsection (Line 326), references 1,7,6,33 are review articles, and Ref. 29 is not a research article. The same was in the other traits as in References 20,32, 34.

we have included the relevant information available for adaptation characteristics from the reviews and unpublished papers and reports which we missed to indicate very well in our search and study inclusion criteria and this portion is included in the methodology section of the review. 

The Objective of the study is improved both the title and the objectives of the study both in abstract portion and in the introduction subsection to meet the included qualitative systematic and quantitative meta-reviews.

Responses to general comments 

Figure 2: Corrections are made by including the full names of the abbreviations in the Box plots consistently. 

Line 481: The first paragraph of the “3.3.2, quality control is removed. Information given under these subsections were moved to the materials and method subsection.

All typographic errors were corrected line by line as indicated in the final version of the paper.

---

## [Editor Report · Decision Letter 1]

12 May 2021

Production, Reproduction and Some Adaptation characteristics of Boran cattle breed under changing climate: A systematic review and meta-analysis

PONE-D-20-39503R1

Dear Dr. Bayssa,

We’re pleased to inform you that your manuscript has been judged scientifically suitable for publication and will be formally accepted for publication once it meets all outstanding technical requirements.

Kind regards,

Dawit Tesfaye

Academic Editor

PLOS ONE
---

## [Editor Report · Acceptance letter]

18 May 2021

PONE-D-20-39503R1 

Production, reproduction and Some adaptation characteristics of Boran cattle breed under changing climate: A systematic review and meta-analysis 

Dear Dr. Bayssa:

I'm pleased to inform you that your manuscript has been deemed suitable for publication in PLOS ONE. Congratulations! Your manuscript is now with our production department. 

Kind regards, 

on behalf of

Dr. Dawit Tesfaye 

Academic Editor

PLOS ONE